# Management practice, quality of life and associated factors in psoriasis patients attending a dermatological center in Ethiopia

Seefu Megarsa Kumsa◉, Tamrat Assefa Tadesse◉*◉, Minyahil Alebachew Woldu◉

Department of Pharmacology and Clinical Pharmacy, School of Pharmacy, College of Health Sciences, Addis Ababa University, Addis Ababa, Ethiopia

◉ These authors contributed equally to this work.
* tamrat.assefa@aau.edu.et

**Data Availability Statement:** All relevant data are within the manuscript.

**Funding:** The authors received no specific funding for this work.

## Abstract

### Background

Psoriasis is a chronic inflammatory disease characterized by keratinocyte hyperproliferation and aberrant differentiation with great negative impact on patients' quality of life (QoL). This study aimed at assessing factors influencing management practice, and QoL and its associated factors among ambulatory psoriatic patients visiting All Africa Leprosy, Tuberculosis and Rehabilitation Training (ALERT) Center in Addis Ababa, Ethiopia.

### Materials and methods

A cross sectional study was conducted in 207 patients with psoriasis attending the dermatology clinic of ALERT Center in Addis Ababa, Ethiopia. Data were collected using structured questionnaire and patients' chart review. Dermatology Life Quality Index (DLQI) was used to measure patients' QoL. Patients' characteristics were summarized using descriptive statistics and predictors of QoL were identified by binary logistic regression.

### Results

Among 207 study participants, 122 (58.9%) were females. The mean age of the study population was 37.92 (SD = 14.86) years (ranging from 16 to 68 years). The mean age at which diagnosis of psoriasis made was 32 (SD = 13.7) years ranging from 10 to 62 years. The duration of the disease in 112 (54.1%) patients were more than or equal to 5 years. Majority of study participants 145 (70.0%) had plaque psoriasis followed by sebopsoriasis, 24 (11.6%). The majority of plaque psoriasis (80%) cases were managed by topical corticosteroids with or without salicylic acid or coal tar and only 21 (14.5%) treated by methotrexate alone. The mean DLQI was 6.25 corresponding to a moderate effect. Symptoms and feelings were the most affected domains of QoL. Factors associated with poor QoL were female [AOR = 0.17 (95%CI: 0.06, 0.48)], low, above average and high family income ([AOR = 0.12 (95% CI: 0.02, 0.56)], [AOR = 0.06 (95% CI:0.01, 0.32)], and [AOR = 0.03 (95% CI: 0.01, 0.22)]), respectively, and primary education level [AOR = 0.14 (95% CI: 0.03, 0.64)] while

**Competing interests:** The authors have declared that no competing interests exist.

**Abbreviations:** AOR, Adjusted Odds Ratio; ALERT, All Africa Leprosy Tuberculosis and Rehabilitation Training; DLQI, Dermatology life quality index; QoL, Quality of life; SPSS, Statistical Package for Social Sciences (SPSS).

being on systemic therapy [AOR = 4.26 (CI: 1.18, 15.35)] was predictor of better QoL. Poor QoL was predominant in females [AOR = 0.17 (95%CI: 0.06, 0.48)], low income [AOR = 0.12 (95% CI: 0.02, 0.56] patients, and patients with primary education level [AOR = 0.14 (95% CI: 0.03, 0.64)]. Patients on systemic therapy [AOR = 4.26 (CI: 1.18, 15.35)] had good QoL.

## Conclusion

Our study identified that topical corticosteroids were the mainstay of psoriasis treatment in the dermatology clinic of ALERT Center in Addis Ababa, Ethiopia. Moderate effect QoL was achieved by study participants based on DLQL score.

## Introduction

Psoriasis is a chronic inflammatory disease affecting skin and joints characterized by erythematous papules and plaques [1]. It is also associated with an array of significant medical and psychiatric comorbidities, including psoriatic arthritis, cardiovascular disease, diabetes, malignancy, depression, and anxiety [2, 3]. Thus, psoriasis requires a timely management otherwise it is subjected to flares and remissions that come in many different variations and degrees of severity which may be symptomatic throughout life and may be progressive with age or wax and wane in its severity [4].

Based on morphologic characteristics, several subtypes of psoriasis have been identified which is necessary to guide appropriate management. However, clinical findings often fit into more than one category. The most common subtypes are plaque, guttate, erythrodermic, pustular, and inverse psoriasis [5, 6]. Psoriasis affects about 2%–3% of the world's population although its prevalence varies among different ethnicities and geographic locations [7]. Higher altitudes are associated with the largest prevalence. Northern Europe and East Asia have the highest and lowest prevalence rates, respectively. Similarly, global reports on psoriasis showed a wide range of prevalence varying from 0.09% to 11.4% [7, 8]. In Africa, epidemiology of psoriasis is not well described. However, the prevalence rates vary from 1.9% to 2.5%, and 0.025% to 0.9% were reported in Eastern and Western Africa countries, respectively [9].

Psoriasis has a negative impact on the psychological status, social interaction, and overall QoL of the patients [10, 11]. Studies showed that psoriasis affects day-to-day activities, occupational, and sexual functioning, often independent of the extent, and severity of skin lesions [11–13]. It was also demonstrated that it can cause anxiety, depression, anger, and embarrassment, which then leads to social isolation and absenteeism at work and school. Social and sporting activities can become difficult since they worry about what other people think of their appearance [14]. It has also great economic and financial consequences [15].

The treatment of psoriasis comprises topical agents, phototherapy, systemic immunosuppressants, and, more recently, biologics [16]. Topical agents are widely used for localized and milder forms of these diseases and to control flares of skin disease in patients with widespread or more severe one. On the other hand, systemic immunosuppressants and biologics are reserved for more severe disease because of side effects associated with chronic use. However, a number of immunobiologics with high efficacy and reduced side effect have recently been introduced [17].

There are unmet needs regarding the management of psoriasis in Africa. A stepwise approach, based on disease severity, is recommended in many African regions for psoriasis

management. When systemic therapy is indicated, methotrexate, cyclosporin A and acitretin are first-line options in Africa countries [18]. Poor multidisciplinary approach, logistic problem of running clinic within outpatient setting, cost of medications, lack of patients understanding of their disease and, poor adherence to medications, limited availability of interested dermatologists and specialist and specialist nurses are limiting factors to provide comprehensive care for psoriatic patients in Africa and other similar developing countries [18]. In Ethiopia, topical agents (steroids, salicylic acid and coal tar), methotrexate, other systemic immunosuppressants and sun are the conventional treatments for psoriasis [9].

There is no generally accepted consensus to declare the treatment of psoriasis is successful or not, however, patient care should include the provision of effective, convenient, and safe drugs, patient-reported outcomes like treatment preference, satisfaction, and QoL [19]. In Ethiopia, very little is known about the level of QoL and its determinants in psoriatic patients. Therefore, this study aimed to evaluate management practice, QoL and its associated factors among ambulatory psoriatic patients attending ALERT Hospital.

## Materials and methods

### Study setting

The study was conducted in the dermatology clinic of All Africa Leprosy, Tuberculosis and Rehabilitation Training Center (ALERT), located in Addis Ababa, Ethiopia. ALERT Center is one of the leading specialized hospitals in dermatology, ophthalmology, and plastic and reconstructive surgery in the country.

### Study design and period

Hospital-based cross-sectional study design that involved interviewing patients and retrospective reviewing their charts was used in this study. Patients' interview and data extraction were conducted from July 01, 2019 to December 31, 2019 and December 31, 2018 to December 31, 2019, respectively.

### Eligibility criteria

**Inclusion criteria.** Patients included in this study were all patients with psoriasis and visiting dermatologic clinic during the study period, and patients with age $\geq$16 years that were on active follow-up and receiving treatment for at least 6 months.

**Exclusion criteria.** Patients who were physically ill, did not answer at least two questions in DLQI questionnaires, and unable to speak Amharic or didn't have a caregiver to facilitate communication were excluded from the study. Moreover, incomplete patients' medical charts were not included in the analysis.

### Sample size and sampling technique

The sample size was computed based on a single population proportion formula. Since there was no previous study conducted on the QoL among psoriasis in Ethiopia, the proportion was taken as 50% and accordingly sample size was calculated to be 384. However, 504 patients ($<$10,000) visited the clinic during the study period and thus the finite population correction formula is used to calculate the actual sample size which was 218. Taking a 10% contingency (non-response rate), the final sample became 240 in our study. A systematic random sampling method was used to recruit samples for the study on each day of the data collection process. Sampling interval ($k^{th}$) was determined by dividing the total number of psoriatic patients that came within the study period by estimated sample size ($k^{th}$ = 504/240 = 2). Thus, the first

patient was selected randomly then every other patient was interviewed following physicians visit until the required sample was reached. In this study, 207 study participants were included in the final analysis after excluding patients with incomplete data.

## Study variables

Dependent variables were management practice and QoL while independent variables were age, sex, educational status, marital status, occupation, monthly income, social drug habits, comorbidity, type of psoriasis, treatment modality, duration of psoriasis treatment, type and number of drugs.

## Data collection tool

Data were collected using a structured data abstraction tool through interviewing patients and reviewing their medical records. The first part of the tool comprised socio-demographic characters (age, sex, educational status, monthly income, occupation, alcohol intake, chat chewing, and smoking status). The second part was the DLQI questionnaire, which was used to measure QoL. The final section comprised a tool to collect relevant clinical data like types of psoriasis, age at initial diagnosis, duration of disease, current medications, and comorbidities. The first part was translated into Amharic language and the Amharic version was administered to study participants.

The DLQI questionnaire was the first dermatology-specific QoL questionnaire. It consists of questions concerning patients' perception of the impact of skin diseases on different aspects of their health-related quality of life over the last week. It was widely used in clinical studies of more than 80 countries and translated into more than 110 languages including Amharic [20]. DLQI is a 10-item questionnaire where question 1 and 2 assess symptoms and feelings; 3 and 4 assess daily activities; 5 and 6, leisure; 7, work or school; 8 and 9, personal relationships; and 10, treatment. DLQI is a 10-item questionnaire assessing patient's symptoms and feelings, daily and leisure activities, work or school status, personal relationships and treatment.

The DLQI was rated on a 4-point scale with corresponding values of 3 = very much, 2 = a lot, 1 = a little and 0 = not at all; 0 = not relevant for each item. The highest sum is 30, and the lowest 0, and interpretation is: 0 to1 score = no effect (normal DLQL); 2 to 5 = small effect; 6 to 10 = moderate effect; 11 to 20 = large effect and 21 to 30 = extremely serious effect on patient's life. DLQI scores > 5 were considered moderate-severely affected the QoL indicating poor QoL and scores ≤ 5 were considered the less affected QoL signifying a better QoL [21].

## Data collectors and quality assurance

Data were collected by two nurses working in the outpatient dermatologic clinic under the supervision of the principal investigator. One-day training was given for data collectors on the study's purpose, how to conduct a patient interview, and collect data from the patients' charts. A pretest was done on twenty patients before actual data collection to check uniformity and understandability of the data collection tool and necessary modifications were made to the data collection tool. The principal investigator closely supervised the data collection process and was giving feedback and correction daily to maintain the quality of data.

## Data analysis

After cleaning incomplete data, collected data was entered and analyzed using Statistical Package for Social Sciences (SPSS) version 25. Descriptive statistics were used for analyses of socio-demographic variables and relevant clinical data. Categorical variables were described by

frequencies and percentages, and continuous variables were expressed by means and standard deviations. Binary logistic regression analysis was used to identify the determinants of QoL. Variables with p<0.25 in univariate analysis were further analyzed by multiple logistic regressions to avoid confounders. A p-value < 0.05 was used to confirm an association.

### Ethical considerations

Ethical approval was received from the Ethical Review Board of the School of Pharmacy, College of Health Sciences, Addis Ababa University (Ref No: ERB/SOP/103/06/2019), and then permission was also obtained from ALERT Hospital, Department of Dermatology. In addition, informed written consent was obtained from the study participants and confidentiality was maintained by omitting patient identifiers and giving code number.

## Results

### Demographic characteristics of patients

A total number of 207 participants were included for the final analysis. 122 (58.9%) were females. The mean age of the study population was 37.92 (SD = 14.86) years (Table 1).

In this study, 124 (59.9%) were married participants and 78 (37.7%) participants had higher education (diploma and above). Of the total patients, 53 (25.6%) were employees, 46 (22.2%) were merchants/self-employed, and 77 (37.2%) patients had a high monthly family income.

### Clinical characteristics of patients

As shown in Table 2, the mean age at which the diagnosis of psoriasis made was 32.0 (SD = 13.7) years. The duration of the disease in 112 (54.1%) patients were more than or equal to 5 years.

Plaque psoriasis was the major clinical characteristics manifested in 145 (70.0%) patients and followed by sebopsoriasis manifestations 24 (11.6%). In the present study, 158 (76.3%) patients were in the normal body mass index (BMI) range and only 6 (2.9%) patients had comorbid diseases among which human immunodeficiency virus (HIV) and hypertension accounted for 1.4% and 1.0%, respectively.

### Anti-psoriatic drug utilization practice

In our study, 185 (89.4%) of patients received topical corticosteroids for the treatment of psoriasis, and 12 (6.5%) patients have been prescribed two topical steroids with different potency. Methotrexate was prescribed for 33 (15.9%) patients. The average number of antipsoriatic drugs prescribed per patient was 1.8 (SD = 0.7). Plaque psoriasis was mostly managed by topical therapy alone, topical corticosteroids with or without salicylic acid/coal tar, 113 (77.9%) but only 21 (14.5%) treated by methotrexate alone. All patients with palmoplantar, sebopsoriasis, scalp, and pustular psoriasis were treated only by topical corticosteroids with or without salicylic acid (Table 3).

### Quality of life domains

According to DLQI, 104 (50.2%) patients experienced a small effect and only 4 (2.0%) patients had an extremely large effect on the QoL (Fig 1).

DLQI domains were affected differently among psoriatic patients and the overall effect in descending order was patient's symptoms and feeling (with a total value of 541), treatment (with a total value of 287), daily activities (a total value of 232), and work and school (a total

**Table 1. Socio-demographic characteristics of psoriasis patients attending dermatology clinic of ALERT Center, Addis Ababa, Ethiopia (N = 207).**

| Socio-demographic | Category | Number (%) |
|---|---|---|
| Sex | Male | 85 (41.1) |
| | Female | 122 (58.9) |
| Age (in years) | 16–35 | 106 (51.2) |
| | 36–60 | 75 (36.2) |
| | >60 | 26 (12.6) |
| Marital status | Single | 61 (29.5) |
| | Married | 124 (59.9) |
| | Divorced | 6 (2.9) |
| | Widowed | 16 (7.7) |
| Educational status | Unable to read and write | 39 (18.8) |
| | Primary school (Grade 1–8) | 46 (22.2) |
| | Secondary school (Grade 9–12) | 44 (21.3) |
| | Higher education (Diploma and above) | 78 (37.7) |
| Occupation | Farmer | 24 (11.6) |
| | Merchant/self-employed | 46 (22.2) |
| | Employee | 53 (25.6) |
| | Unemployed | 14 (6.8) |
| | Housewife | 22 (10.6) |
| | Student | 29 (14.0) |
| | Daily laborer | 10 (4.8) |
| | Others* | 9 (4.3) |
| Monthly family income (ETB)** | Very low (≤860) | 32 (15.5) |
| | Low (861–1500) | 42 (20.3) |
| | Average (1501–3000) | 24 (11.6) |
| | Above average (3001–5000) | 32 (15.5) |
| | High (≥5001) | 77 (37.2) |
| Social drug habits | Active cigarette smoker | 3 (1.4) |
| | Regular alcohol user | 7 (3.4) |
| | Regular chat chewer | 11 (5.3) |

ETB- Ethiopian birr

*retired, broker, guard

** based on Ethiopian civil service monthly salary scale for civil servants.

value of 54 (Table 4). The mean DLQI was 6.25 corresponding to a moderate effect and patient's symptoms and feeling were the most affected domains of QoL.

## Factors associated with quality of life

Based on the results of univariate binary logistic regression analysis, variables such as gender, age group, educational status, occupation, family income, regular use of alcohol, BMI, age at initial diagnosis, and treatment modality were included in the multivariate binary logistic regression analysis. Accordingly, gender, educational status, family income, and treatment modality had a significant association with quality of life. The odds of having better QoL (DLQI≤5) for female patients was decreased by 83% [AOR = 0.17 (95% CI: 0.06, 0.48)], (P = 0.001). This study also found a significant association between primary educational status and poor quality of life [AOR = 0.14 (95% CI: 0.03, 0.64)] (P = 0.011). The odds of having

**Table 2. Clinical characteristics of psoriasis patient attending dermatology clinic of ALERT Center, Addis Ababa, Ethiopia (N = 207).**

| Factors | Category | N (%) | Mean± SD | Range |
|---|---|---|---|---|
| BMI (Kg/m$^2$) | ≤18.5 | 15 (7.2) | | |
| | 18.5–24.9 | 158 (76.3) | | |
| | 25.0–29.9 | 17 (8.2) | | |
| Age at initial diagnosis (in years) | Early onset (<40) | 144 (69.6) | 32.0±13.7 | 10.0–62.0 |
| | Late-onset (≥40) | 63 (30.4) | | |
| Duration of the disease (in years) | <5 | 95 (45.9) | 5.6±3.6 | 0.8–18.7 |
| | ≥ 5 | 112 (54.1) | | |
| Types of psoriasis | Plaque psoriasis | 145 (70.0) | | |
| | Sebopsoriasis | 24 (11.6) | | |
| | Scalp psoriasis | 22 (10.6) | | |
| | Palmoplantar psoriasis | 13 (6.3) | | |
| | Pustular psoriasis | 2 (1.0) | | |
| | Inverse and nail psoriasis | 1 (0.5) | | |
| Comorbid diseases | Yes | 6 (2.9) | | |
| | No | 201 (97.1) | | |
| Specific comorbidity presented | HIV | 3 (1.4) | | |
| | Hypertension | 2 (1.0) | | |
| | Diabetes mellitus and hypertension | 1 (0.5) | | |
| Treatment modality | Topical corticosteroids | 174 (84.1) | | |
| | Systemic agent | 22 (10.6) | | |
| | Topical corticosteroids and systemic agents | 11 (5.3) | | |

better QoL (DLQI≤5) were more than four times higher among patients on systemic therapy as compared to those on topical only [AOR = 4.26 (CI:1.18,15.35)] (P = 0.027) (Table 5).

## Discussion

In this study, different medication regimens were used in the management of psoriasis. Topical corticosteroids were the most commonly prescribed medication followed by methotrexate in

**Table 3. Anti-psoriatic drugs utilization pattern among psoriasis patients attending ALERT Center, Addis Ababa, Ethiopia (N = 207).**

| Treatments | Types of Psoriasis, N (%) | | | | | | Total |
|---|---|---|---|---|---|---|---|
| | Plaque | Inverse and nail | Palmoplantar | Sebopsoriasis | Scalp | Pustular | |
| | N = 145 | N = 1 | N = 11 | N = 26 | N = 22 | N = 2 | N (%) |
| **Topical corticosteroids** | | | | | | | 185 (89.4) |
| Betamethasone dipropoinate | 106 (73.1) | 0 (0) | 2 (18.2) | 22 (84.6) | 20 (90.9) | 2 (100) | 150 (81.1) |
| Clobetasone propionate | 5 (3.4) | 0 (0) | 6 (54.5) | 0 (0) | 0 (0) | 0 (0) | 12 (6.5) |
| Mometasone furoate | 11 (7.6) | 0 (0) | 2 (18.2) | 4 (16.7) | 0 (0) | 0 (0) | 17 (9.2) |
| Clocortolone pivolate | 8 (5.5) | 0 (0) | 3 (27.3) | 0 (0) | 2 (9.1) | 0 (0) | 12 (6.5) |
| **Other topical agents** | | | | | | | |
| Salicylic acid | 86 (59.3) | 0 (0) | 7 (63.6) | 16 (66.7) | 6 (27.3) | 0 (0) | 115 (55.6) |
| Coal tar | 8 (5.5) | 0 (0) | 0 (0) | 0 (0) | 0 (0) | 0 (0) | 8 (3.9) |
| **Systemic agents (traditional)** | | | | | | | |
| Methotrexate | 32 (22.1) | 1 (100) | 0 (0) | 0 (0) | 0 (0) | 0 (0) | 33 (16.0) |
| **Mean ± SD** | | | | | | | |
| No. of drugs | 1.8±0.7 | 1.0±0.0 | 2.0±0.7 | 2.5± 0.7 | 1.4±0.5 | 1.0±0.0 | 1.8±0.7 |
| No. of preparation | 1.2±0.4 | 1.0±0.0 | 1.2±0.4 | 1.7±0.5 | 1.1±0.3 | 1.0±0.0 | 1.2±0.4 |

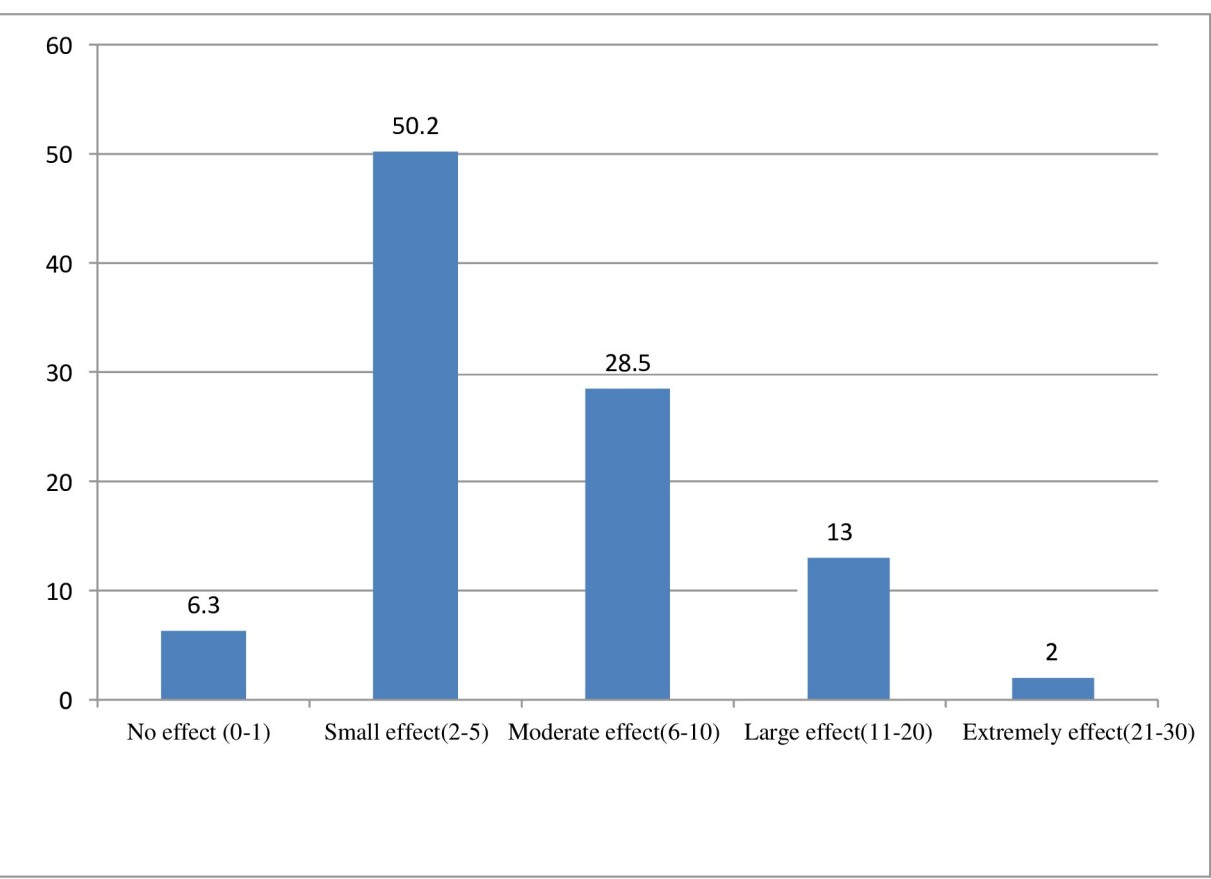

**Fig 1. Proportion of overall effect of psoriatic on QoL among psoriasis patients attending ALERT Hospital, Addis Ababa, Ethiopia (N = 207).**

treating plaque psoriasis Likewise, different guidelines regarding the treatment of psoriasis vulgaris have shown that topical corticosteroids and methotrexate are one of the first lines for treating mild and moderate-severe psoriasis, respectively [21–23]. Corticosteroids can even be used as an adjunct to systemic therapy or phototherapy for moderate-to-severe psoriasis [5,

**Table 4. Quality of life (QoL) domains of psoriatic patients attending dermatologic clinic of ALERT Hospital, Addis Ababa, Ethiopia (N = 207).**

| Domains | Question Number | Mean | Minimum value | Maximum value | Total value |
|---|---|---|---|---|---|
| Symptoms and feelings | 1 | 1.84 | 0 | 3 | 380 |
|  | 2 | 0.78 | 0 | 3 | 161 |
| Daily activities | 3 | 0.22 | 0 | 3 | 45 |
|  | 4 | 0.90 | 0 | 3 | 187 |
| Leisure | 5 | 0.23 | 0 | 3 | 47 |
|  | 6 | 0.22 | 0 | 3 | 46 |
| Work and School | 7 | 0.25 | 0 | 3 | 51 |
|  |  | 0.02 | 0 | 1 | 3 |
| Personal relationships | 8 | 0.31 | 0 | 3 | 64 |
|  | 9 | 0.11 | 0 | 3 | 23 |
| Treatment | 10 | 1.39 | 0 | 3 | 287 |
| **Overall DLQI** |  | 6.25 | 0 | 29 | 1294 |

**Table 5. Logistic regression analysis of factors associated with QoL among psoriasis patients attending dermatologic clinic of ALERT Hospital, Addis Ababa, Ethiopia (N = 207).**

| Variables | DLQI, n (%) | | OR (95% CI) | | P-value |
|---|---|---|---|---|---|
| | ≤5 | >5 | Crude | Adjusted | |
| **Sex** | | | | | |
| Male | 35 (41.18) | 50 (58.82) | 1.00 | 1.00 | |
| Female | 82 (67.21) | 40 (32.79) | 0.34 (0.19,0.61) | 0.17 (0.06,0.48) | 0.001* |
| **Age group** | | | | | |
| 16–35 | 66 (62.26) | 40 (37.74) | 1.00 | 1.00 | |
| 36–65 | 44 (58.67) | 31 (41.33) | 1.16 (0.64,2.13) | 0.92 (0.33,2.60) | 0.872 |
| >65 | 7 (26.92) | 19 (73.08) | 4.48 (1.73,11.60) | 5.50 (0.68,44.27) | 0.109 |
| **Educational status** | | | | | |
| Can't read and write | 18 (46.15) | 21 (53.85) | 1.00 | 1.00 | |
| Primary | 38 (82.61) | 8 (17.39) | 0.18 (0.07,0.49) | 0.14 (0.03,0.64) | 0.011* |
| Secondary | 22 (50) | 22 (50) | 0.86 (0.36,2.03) | 3.26 (0.65,16.50) | 0.152 |
| Higher | 39 (50) | 39 (50) | 0.86 (0.40,1.85) | 4.18 (0.75,23.62) | 0.102 |
| **Occupation** | | | | | |
| Farmer | 11 (45.83) | 13 (54.17) | 1.00 | 1.00 | |
| Merchant | 33 (71.74) | 13 (28.26) | 0.33 (0.12,0.93) | 0.87 (0.16,4.56) | 0.865 |
| Employee | 27 (50.94) | 26 (49.06) | 0.82 (0.31,2.14) | 1.28 (0.23,7.07) | 0.774 |
| Unemployed | 5 (35.71) | 9 (64.29) | 1.52 (0.39,5.91) | 1.89 (0.22,16.00) | 0.560 |
| House wife | 14 (63.64) | 8 (36.36) | 0.48 (0.15,1.58) | 4.80 (0.78,29.60) | 0.091 |
| Student | 21 (72.41) | 8 (27.59) | 0.32 (0.10,1.01) | 2.32 (0.26,20.80) | 0.450 |
| Daily laborer | 4 (40) | 6 (60) | 1.27 (0.28,5.68) | 5.48 (0.56,53.95) | 0.145 |
| Others | 2 (22.22) | 7 (77.78) | 2.96 (0.51,17.30) | 0.94 (0.04,20.11) | 0.970 |
| **Monthly family income (ETB)** | | | | | |
| Very low ≤860) | 10 (31.25) | 22 (68.75) | 1.00 | 1.00 | |
| Low (861–1500) | 32 (76.19) | 10 (23.81) | 0.14 (0.05,0.40) | 0.12 (0.02,0.56) | 0.007* |
| Average (1501–3000) | 8 (33.33) | 16 (67.67) | 0.91 (0.29,2.82) | 0.24 (0.04,1.43) | 0.117 |
| Above average (3001–5000) | 19 (59.38) | 13 (40.62) | 0.31 (0.11,0.87) | 0.06 (0.01,0.32) | 0.001* |
| High (≥5001) | 48 (62.34) | 29 (37.66) | 0.28 (0.11,0.66) | 0.03 (0.01,0.22) | <0.001* |
| **Regular use of alcohol** | | | | | |
| No | 115 (57.79) | 84 (42.21) | 1.00 | 1.00 | |
| Yes | 2 (25) | 6 (75) | 4.11 (0.81,20.85) | 8.05 (0.55,117.65) | 0.128 |
| **BMI** | | | | | |
| Underweight | 3 (20) | 12 (80) | 1.00 | 1.00 | |
| Normal | 93 (58.86) | 65 (41.14) | 0.18 (0.05,0.64) | 0.28 (0.06,1.36) | 0.114 |
| Overweight | 7 (41.18) | 10 (58.82) | 0.36 (0.07,1.75) | 0.86 (0.10,7.31) | 0.889 |
| **Age at initial diagnosis (in years)** | | | | | |
| <40 | 90 (62.5) | 54 (37.5) | 1.00 | 1.00 | |
| ≥ 40 | 27 (42.86) | 36 (57.14) | 2.22 (1.22,4.06) | 1.82 (0.47,7.04) | 0.384 |
| **Treatment modality** | | | | | |
| Topical | 104 (59.43) | 71 (40.57) | 1.00 | 1.00 | |
| Systemic | 7 (33.33) | 14 (67.67) | 2.93 (1.13,7.62) | 4.26 (1.18,15.35) | 0.027* |
| Topical and systemic | 6 (54.54) | 5 (45.46) | 1.22 (0.36,4.15) | 0.73 (0.10,5.61) | 0.765 |

*- statistically significant at P<0.05.

22] Among patients with plaque psoriasis, extemporaneous preparation of salicylic acid with topical corticosteroids was prescribed for 86 (59.3%) patients. Palmoplantar sebopsoriasis and scalp psoriasis were managed by topical corticosteroids alone or in combination with salicylic acid. This is in line with different clinical guidelines [23, 24].

Regarding the management of pustular psoriasis, the present study revealed that topical corticosteroids were commonly used treatment regimen. Contrastingly, National Psoriasis Foundation clinical guideline [24] did not state topical corticosteroids both as first and second-line therapy for treating such patients but rather recommended biologics. This might be due to unavailability of biologics and lack of local standard treatment guidelines for the management of psoriasis in Ethiopia. Extemporaneous preparation containing coal tar with topical corticosteroids was prescribed for 8 (3.9%) patients. South African guideline also recommends coal tar for management of patient s with mild psoriasis who require an inexpensive alternative to corticosteroids [22]. Furthermore, various guidelines also suggested that coal tar can be used for treating localized psoriasis, but it has poor tolerability by patients due to its irritation property, cosmetic issues like odor, and staining of clothes [5, 23, 25]. So, patients prescribed for coal tar need higher attention.

The present study reported a moderate effect of psoriasis on patients' QoL, which is consistent with a hospital-based cross-sectional study done in India [26], Malaysia [27], and Russia [28]. However, Egyptian study found a higher mean DLQL, corresponding to a very large effect [29]. The first reason for this variation could be our study included all types of psoriasis, unlike the study done in Egypt which incorporated only psoriasis of chronic plaque-type and the second reason might be the difference in the severity of the disease among study participants as far as this study was carried out among outpatients only.

In the present study, only few patients replied that the disease has no effect on their QoL in consistent with a report of Indian study [26]. Among DLQI domains symptoms and feelings were the highly affected domain. The present result is in line with studies done in India [26, 30], Taiwan [31], and Egypt [29]. This is due to the fact that the domain encompasses questions directed to identify the level of embarrassment and painfulness or itchiness owing to the disease which is used to even define the disease in the case of global reports on psoriasis [8].

The identification of risk factors for a marked reduction in QoL owing to psoriasis is important as it helps to identify the most susceptible patient who requires close monitoring. The result of this study showed female gender was found to be independent predictors, which increases the chance of having poor QoL. This was supported by similar studies done in India [32], the United States of America [33], and Spain [34]. The probable reason could be the fact that females are more concerned about their body image and physical appearance than their male counterparts and they are more likely to feel troubled in public settings.

In this study, patients with primary education status, were significantly associated with poor QoL that is in contrary to a study done in India [35]. Such discrepancy might be a result of a difference in cultural settings, adjustment of confounding factors, DLQI cut-off point, and analysis model. Patients taking systemic therapy were significantly associated with less affected QoL which might be owing to its more aggressive, comfortable, and less time-consuming nature as compared to topical therapy. This result is supported by findings from a cohort study in Spain [34].

Our study had some limitations. The temporal relationship between the dependent and independent variables does not be allowed due to cross-sectional nature of the study design. The other limitation is that the drugs considered during analysis were those prescribed at the time of recent follow-up; it didn't include drugs that were switched or withdrawn. This study also considered only pharmacologic interventions due to poor recording practice of the hospital.

## Conclusions

Topical corticosteroids were the most prescribed treatment modality for psoriasis. Psoriasis moderately affected the QoL patients, patient's symptoms and feelings being the most highly affected domain.

## Supporting information

**S1 Questionnaire. Questionnaire for patient interviews.**
(DOCX)

**S2 Questionnaire. Amharic version questionnaire.**
(DOCX)

**S1 File. Data abstraction format for reviewing patient medical records.**
(DOCX)

## Acknowledgments

We acknowledge ALERT Center for allowing us to conduct this study. The authors also would like to acknowledge all participants, data collectors, staff working in the dermatologic clinic of ALERT Center for their time, and voluntary facilitation for the data collection process.

## Author Contributions

**Conceptualization:** Seefu Megarsa Kumsa, Tamrat Assefa Tadesse, Minyahil Alebachew Woldu.

**Data curation:** Seefu Megarsa Kumsa.

**Formal analysis:** Seefu Megarsa Kumsa.

**Investigation:** Seefu Megarsa Kumsa, Tamrat Assefa Tadesse, Minyahil Alebachew Woldu.

**Supervision:** Tamrat Assefa Tadesse, Minyahil Alebachew Woldu.

**Writing – original draft:** Seefu Megarsa Kumsa.

**Writing – review & editing:** Tamrat Assefa Tadesse, Minyahil Alebachew Woldu.

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
