## [Decision Letter · Decision Letter 0]

21 Jul 2021

PONE-D-20-33787

Management practice, and quality of life and its associated factors among ambulatory psoriatic patients attending Dermatological Center in Addis Ababa, Ethiopia

PLOS ONE

Dear Dr. Tadesse,

Thank you for submitting your manuscript to PLOS ONE. After careful consideration, we feel that it has merit but does not fully meet PLOS ONE’s publication criteria as it currently stands. Therefore, we invite you to submit a revised version of the manuscript that addresses the points raised during the review process.

We look forward to receiving your revised manuscript.

Kind regards,

Filipe Prazeres, MD, MSc, Ph.D.

Academic Editor

PLOS ONE

Journal Requirements:

A clean copy of the edited manuscript (uploaded as the new *manuscript* file).

3. Please include additional information regarding the survey or questionnaire used in the study and ensure that you have provided sufficient details that others could replicate the analyses. For instance, if you developed a questionnaire as part of this study and it is not under a copyright more restrictive than CC-BY, please include a copy, in both the original language and English, as Supporting Information, or include a citation if it has been published previously.

5. We note you have included a table to which you do not refer in the text of your manuscript. Please ensure that you refer to Table 3 in your text; if accepted, production will need this reference to link the reader to the Table.

Reviewers' comments:

Reviewer's Responses to Questions

**Comments to the Author**

1. Is the manuscript technically sound, and do the data support the conclusions?

Reviewer #1: Partly

Reviewer #2: Partly

2. Has the statistical analysis been performed appropriately and rigorously? 

Reviewer #1: Yes

Reviewer #2: Yes

3. Have the authors made all data underlying the findings in their manuscript fully available?

Reviewer #1: Yes

Reviewer #2: Yes

4. Is the manuscript presented in an intelligible fashion and written in standard English?

Reviewer #1: Yes

Reviewer #2: Yes

5. Review Comments to the Author

Reviewer #1: In this manuscript for a research article, authors investigated the clinical care, quality of life and various associated factors of psoriatic patients managed at an outpatient clinic located in Addis Ababa, Ethiopia. As previous data published on the management of psoriasis in East Africa is very limited, the topic and findings of this paper is interesting. However, there are some major issues to be clarified by the authors.

Major remarks:

1. The exact inclusion and exclusion criteria of this study should be described instead of just reporting general eligibility criteria.

2. It is very surprising data that only 2.9% of the patients had comorbid diseases. Were the general medical records of the patients also reviewed? These data should be also discussed.

3. It should be described in the Introduction, which topicals, drugs and biologics are available for the treatment of psoriasis in Ethiopia, as treatment choices also highly depends on the availability of medications.

4. In the Discussion, it its not necessary to describe how corticosteroids and methotrexate work but would be important to compare the used treatment modalities to those of other regions and countries. Also, the benefits and limitations of inexpensive treatment options such as coal tar ointments should be discussed.

Minor remarks:

1. In the abstract, it is not necessary to include technical details, such as the name of the exact statistical software used for data analysis.

2. In the Introduction, epidemiologic data of psoriasis is only mentioned for Asia and Europe. It would be more appropriate to describe epidemiologic data from Africa. Also, previous papers describing the care for psoriatic patients in East Africa should be cited here.

3. In the Study setting section, it is confusing that only some general facts on the department, such as when it was established and in which medical specialties it is a leading center is reported. The prospective nature of the study should be described here.

4. The price of each treatment option should be also discussed in addition to their benefit in QOL, especially as the income of the patients was also reported.

5. Further conclusions should be added based on the observations of this study instead of repetition of the results.

Reviewer #2: The data support the conclusions, but for some of the statistical analyses, groups selected for O.R. 1.00 should be groups with a higher number of subjects for to increase the reliability of the results, and possibly avoiding errors, as mentioned in my comments. It is not likely, though, that this improvement will change the conclusions.

This is an interesting, informative and useful article, but it needs some changes.

Major concern:

When comparing groups, the largest group should be the basic group to compare with. Special caution is important when the reference group, with an O.R. of 1.00, is small. The result of switching to the largest group or the largest group at another end of a scale (for instance high income vs. low income) may change results and conclusions, although it is not likely that the main conclusions will be changed.

Reference to the first article from 1994 about DLQI values in healthy subjects, or another article including control or healthy subjects, should have been described explicitly, but briefly, with some numbers. What is a “normal” DLQI?

Preferable changes:

In the main text or as supportive information or footnotes, the variables used for adjustment should be indicated. If all variables of the independent ones is used except for the result variable (not preferable), this should be indicated. Huber’s stepdown procedure, letting age and gender remain as correction factors, is a feasible solution. If you adjust for non-significant variables, important results may be lost. However, age group and gender should always be adjusted for. However, using p<0.25 for selecting initial variables for adjustment, as you did, is common and preferable.

The distribution of results, as with spreadsheets or histograms, should have be shown. These findings could also be included as supportive information online. This way, one could see that the work is properly done without being obliged to read everything.

Minor concerns:

See some of the remarks added to the text file.

Recommendation in the case of further studies:

When dealing with data with a continuous outcome or more than two outcomes, ordinary regression, or, if the outcome is not normally distributed, and also appropriate if the data are normally Distributed, robust regression is preferable.

6. PLOS authors have the option to publish the peer review history of their article (what does this mean?). If published, this will include your full peer review and any attached files.

Reviewer #1: No

Reviewer #2: **Yes: **Leif Bjarte Rolfsjord

---

## [Author Response · Author response to Decision Letter 0]

13 Oct 2021

First of all, the authors would like to greatly appreciate and respect for the time and efforts of reviewers of this manuscript for the careful reading and comments on the manuscript, which helped us to improve the manuscript. We have seen all comments carefully and explained at our level best in point-wise manner and incorporate the accepted comments in the revised version of the manuscript. We have given response to the comments made by the editor and reviewer(s) as follows. We tried to prepare the revised manuscript based on PLOS ONE's style requirements.

Point by point responses to reviewers’ comments:

REVIEWER 1

Major remarks:

1. The exact inclusion and exclusion criteria of this study should be described instead of just reporting general eligibility criteria.

Response: We accept comment provided from the reviewer and included inclusion and exclusion criteria in the revised manuscript as suggested by the reviewer

2. It is very surprising data that only 2.9% of the patients had comorbid diseases. Wre the general medical records of the patients also reviewed? These data should be also discussed.

Response: All medical records of the patients were reviewed and we also discussed the data. Since these patient come to the center manly for dermatological conditions management, they may have follow ups at other health facilities in the country. We included all co-morbidities we obtained from our review 

3. It should be described in the Introduction, which topicals, drugs and biologics are available for the treatment of psoriasis in Ethiopia, as treatment choices also highly depends on the availability of medications.

Response: We accept this comment, included treatment options for psoriasis in Ethiopia.

In the Discussion, it is not necessary to describe how corticosteroids and methotrexate work but would be important to compare the used treatment modalities to those of other regions and countries. 

4. Response: We accept this comment and deleted statements which described about how corticosteroids and methotrexate work. Furthermore, we tried to compare, the used treatment modalities to those of other regions and countries as indicated on the first paragraph of discussion.

Also, the benefits and limitations of inexpensive treatment options such as coal tar ointments should be discussed.

Response: We included about this on first paragraph of discussion as indicated below

Guideline on the management of psoriasis in South Africa showed that coal tar is recommended in patients with mild psoriasis who require an inexpensive alternative to corticosteroids [21]. Various guidelines also exposed that coal tar can be used for treating localized psoriasis, but it has poor tolerability by patients due to its irritation property, cosmetic issues like odor, and staining of clothes [5, 19,23]. So, patients prescribed for coal tar need higher attention.

Minor remarks:

1. In the abstract, it is not necessary to include technical details, such as the name of the exact statistical software used for data analysis.

Response: We removed name of the exact statistical software used for data analysis

2. In the Introduction, epidemiologic data of psoriasis is only mentioned for Asia and Europe. It would be more appropriate to describe epidemiologic data from Africa.

Response: We accept this comment, but unable to describe it as there is no epidemiological data of psoriasis in Africa 

Also, previous papers describing the care for psoriatic patients in East Africa should be cited here.

Response: We have included this in the revised manuscript

3. In the Study setting section, it is confusing that only some general facts on the department, such as when it was established and in which medical specialties it is a leading center is reported. The prospective nature of the study should be described here.

Response: We already included about nature of the study under study design and period section.

4. The price of each treatment option should be also discussed in addition to their benefit in QOL, especially as the income of the patients was also reported.

Response: We didn’t discuss the price of the medications as far as it various from time to time and also mostly medications were out of stock at the studied center.

5. Further conclusions should be added based on the observations of this study instead of repetition of the results.

Response: We revised our conclusion both in abstract and main text according to this comment. 

REVIEWER 2

The data support the conclusions, but for some of the statistical analyses, groups selected for O.R. 1.00 should be groups with a higher number of subjects for to increase the reliability of the results, and possibly avoiding errors, as mentioned in my comments. It is not likely, though, that this improvement will change the conclusions.

This is an interesting, informative and useful article, but it needs some changes

Response: We take this recommendation, however from our limited experience in data analysis which need such computing we use both ways i.e. why we did in other way round. 

Major concern

When comparing groups, the largest group should be the basic group to compare with. Special caution is important when the reference group, with an O.R. of 1.00 is small. The result of switching to the largest group at another end of scale (for instance high income vs. low income) may change results and conclusions, although it is not likely that the main conclusions will be changed.

Response: As far as we developed the questionnaire prior to data collection, which we did in ascending order, it is impossible to take the middle as a control group and to do such it requires the data to be entered newly which as finding is no changed. The main thing here is that, knowing the reference and being in caution while interpreting the results. We did this carefully during data analysis. A mentioned by reviewer, the main conclusions will not be changed.

Reference to the first article from 1994 about DLQI values in healthy subjects, or another article including control or healthy subjects, should have been described explicitly, but briefly, with some numbers. What is a ‘normal’ DLQI? 

Response: The normal DLQI means the disease has no impact on the patient, which is DLQI score of 0-1 as also indicated by no effect as shown in our data collection tool section of manuscript .

Preferable changes

In the main text or as supportive information or footnotes, the variables used for adjustment should be indicated. If all variables of the independent ones are used except for the result variable (not preferable), this should be indicated. Huber’s stepdown procedure, letting age and gender remain as correction factors, is a feasible solution. If you adjust for non-significant variables, important results may be lost. However, age group and gender should always be adjusted for. However, using p<0.25 for selecting initial variables for adjustment, as you did, is common and preferable.

Response: We accept this comment and based on that experience we used p<0.25 for selecting initial variables for adjustment. We will take this advice for our future works. Thank you!

The distribution of results, as with spreadsheets or histograms, should have been shown. These findings could also be included as supportive information online. This way, one could see that the work is properly done without being obliged to read everything.

 Response: We used included all results appropriately in this manuscript. We will include supportive information online as supplementary materials)

---

## [Decision Letter · Decision Letter 1]

8 Nov 2021

Management practice, quality of life and associated factors in ambulatory psoriatic patients attending Dermatological Center in Addis Ababa, Ethiopia

PONE-D-20-33787R1

Dear Dr. Tadesse,

We’re pleased to inform you that your manuscript has been judged scientifically suitable for publication and will be formally accepted for publication once it meets all outstanding technical requirements.

Kind regards,

Filipe Prazeres, MD, MSc, Ph.D.

Academic Editor

PLOS ONE

Additional Editor Comments (optional):

Reviewers' comments:

Reviewer's Responses to Questions

**Comments to the Author**

1. If the authors have adequately addressed your comments raised in a previous round of review and you feel that this manuscript is now acceptable for publication, you may indicate that here to bypass the “Comments to the Author” section, enter your conflict of interest statement in the “Confidential to Editor” section, and submit your "Accept" recommendation.

Reviewer #1: All comments have been addressed

2. Is the manuscript technically sound, and do the data support the conclusions?

Reviewer #1: Yes

3. Has the statistical analysis been performed appropriately and rigorously? 

Reviewer #1: Yes

4. Have the authors made all data underlying the findings in their manuscript fully available?

Reviewer #1: No

5. Is the manuscript presented in an intelligible fashion and written in standard English?

Reviewer #1: Yes

6. Review Comments to the Author

Reviewer #1: I thank the authors for carefully addressing all or the raised issues. I find the revised manuscript to show a significant improvement regarding both the presentation and the interpetation of data.

7. PLOS authors have the option to publish the peer review history of their article (what does this mean?). If published, this will include your full peer review and any attached files.

Reviewer #1: No

---

## [Editor Report · Acceptance letter]

10 Nov 2021

PONE-D-20-33787R1 

Management practice, quality of life and associated factors in psoriasis patients attending a Dermatological Center in Ethiopia 

Dear Dr. Tadesse:

I'm pleased to inform you that your manuscript has been deemed suitable for publication in PLOS ONE. Congratulations! Your manuscript is now with our production department. 

Kind regards, 

on behalf of

Prof. Filipe Prazeres 

Academic Editor

PLOS ONE